# The Role of Psychological Factors in Young Adult Snacking: Exploring the Intention–Behaviour Gap

**DOI:** 10.3390/nu17162681

**Published:** 2025-08-19

**Authors:** Astrid Green, Barbara Mullan, Indita Dorina

**Affiliations:** Behavioural Science and Health Research Group, enAble Research Institute, School of Population Health, Faulty of Health Sciences, Curtin University, Bentley, WA 6102, Australia; astrid.green@curtin.edu.au (A.G.); indita.dorina@curtin.edu.au (I.D.)

**Keywords:** unhealthy snacking, dietary decision-making, intention, appetitive traits, enjoyment of food, satiety responsiveness, stress

## Abstract

**Background/Objectives**: Unhealthy snacking, most commonly consisting of sweets, savoury snacks and sugar-sweetened beverages, is associated with various adverse health outcomes. As long-term eating behaviours are commonly established in young adulthood, it is imperative to understand young adults’ dietary decision-making to encourage healthy eating. This study aimed to identify the factors of unhealthy snacking among young adults across the three main snack types. **Methods**: Australian young adults (*N* = 323, *M* = 24.73, *SD* = 3.23) completed an online questionnaire assessing their physical activity engagement, intention, appetitive traits (satiety responsiveness and enjoyment of food), stress and consumption of sweet snacks, savoury snacks and sugar-sweetened beverages. **Results**: Multiple regression analyses indicated that physical activity engagement and stress were significant factors of sweet snack consumption. Physical activity engagement was the only significant factor of savoury snack consumption. Physical activity engagement and satiety responsiveness were significant factors of sugar-sweetened beverage consumption. **Conclusions**: Findings identified factors to bridge the intention–behaviour gap in unhealthy snacking. Results support the evidence that rational dietary decision-making can be interrupted by less conscious cognitive or physiological processes. Interventions should consider the differential importance of factors contributing to the consumption of different snack types to reduce unhealthy snacking.

## 1. Introduction

Unhealthy snacking is a significant public health concern among young adults, with global trends indicating an increase in consumption over the past four decades [1]. Driven by modern urbanisation and changing consumer lifestyles, which prioritise convenient food options, the global snacking industry accounts for over USD $135 billion annually, as of 2024 [2]. In Australia, snack food consumption increased by 10% per capita between 2018 and 2023, with 38.6% of total dietary energy intake derived from snack foods [3]. However, the main source of energy from snacks consumed is from sweets (e.g., lollies, cakes, chocolate), high-sodium or salty snacks (e.g., potato chips, processed meats), and sugar-sweetened beverages [4]. Other healthier snack options, like fruits and vegetables, are consumed less frequently compared to unhealthy snack options [3].

Although a conclusive definition of snacking does not currently exist, it is commonly defined as the consumption of food or drinks between standard mealtimes [5]. Based on existing evidence, general snacking frequency does not directly and reliably predict adverse dietary outcomes such as excess energy intake and adiposity [6]. However, snack food choice is an influential determinant [7]. High consumption of unhealthy snacks, or foods high in sugar, salt and fats, and low in nutrients [1], is associated with adverse health implications. These include the development of chronic and non-communicable diseases such as cardiovascular diseases, Type 2 diabetes, cancers, obesity and depression, among others [1,7,8,9,10,11,12]. Over the lifespan, men are more likely to develop cardiovascular disease [13], stroke and lung disease than women, while women are more likely to develop arthritis [14], depression and anxiety than men [15].

Unhealthy dietary behaviours are especially a concern in young adults, as the short and long-term non-communicable disease risks and outcomes can be influenced by health behaviours established in this transition between adolescence and adulthood [16]. Significant changes occur in young adults’ lives, including starting post-secondary education and moving away from home [17]. During this period, young adults often note obstacles to healthy eating related to inefficient time management, preparation needs, low motivation and greater taste preference for unhealthy foods [18]. For university students, barriers include restricted physical environments (e.g., lack of cooking facilities and food availability), study commitments (e.g., timetables, exams), a lack of self-confidence in nutrition knowledge, and navigating personal factors, including personal, cultural and religious beliefs [19]. Therefore, it is important to better understand the factors that influence unhealthy snacking in young adults to better promote long-term healthy dietary behaviours.

### 1.1. Theory of Planned Behaviour and the Intention–Behaviour Gap

The theory of planned behaviour [20] is commonly applied in psychological and behavioural research to explore the determinants of health behaviours. The theory has been successfully applied to various dietary behaviours, including healthy food intake [21], snacking behaviour [22] and fast food consumption [23]. It was proposed in the theory that intention is the most proximal predictor of behaviour, with stronger intentions to perform a behaviour being more strongly associated with behavioural engagement. Intention is influenced by three main predictors: attitude (beliefs about the potential consequences of behaviour), subjective norms (perception of others’ approval regarding the behaviour) and perceived behavioural control (beliefs about the difficulty of performing the behaviour).

However, positive intentions do not linearly translate into desired behaviour [24]. Although most young adults understand the beneficial health impacts of healthy eating, few report frequently selecting healthy snacks over unhealthy snacks [25]. Meta-analytic findings indicate that the theory may better predict intention than behaviour, with the theory variables being able to account for 39% of variance in intention and 27% of variance in behaviour [26]. Further, Webb and Sheeran [27] demonstrate that medium to large changes in intention (*d* = 0.66) only result in small to medium changes in behaviour (*d* = 0.36). This is known as the intention–behaviour gap and is commonly noted as a weakness of the theory of planned behaviour [28,29]. In subsequent attempts to improve the predictive ability of the theory of planned behaviour, Hall and Fong [29] note that the theory may best predict conscious, controllable and less habitual behaviours. Contrary to the theory of planned behaviour, intention is only likely to directly and primarily influence discrete (one-off) behaviours in relatively supportive ecological contexts. Therefore, other variables may need to be considered when assessing unhealthy snacking, as it is often hedonically motivated, habitual and unintentionally performed due to lapses in self-control [27,30,31].

### 1.2. Factors of Unhealthy Snacking

Although underreported, physical activity can be a confounding factor of high snacking frequency [7]. Individuals who engage in physical activity are more likely to reward themselves with food than individuals who do not [32]. Similarly, individuals who perceive expending more energy may consume more food compared to those who do not [33]. Therefore, it may be beneficial to control for physical activity engagement when exploring the factors of unhealthy snacking.

Other variables that can bridge the intention–behaviour gap in unhealthy snacking include appetitive traits and stress. Appetitive traits are predispositions toward food that influence individuals’ susceptibility to overeating [34]. Having higher food-approach appetitive traits like greater enjoyment of food, and lower food-avoidance appetitive traits like satiety responsiveness, reflect greater appetites and are linked with overweight and obesity in later life [35]. University students with moderate to high anxiety during stressful periods, such as during COVID-19, report greater maladaptive satiety responsiveness, enjoyment of food and emotional over-eating [36].

Generally, studies have linked higher stress with greater unhealthy food consumption [37,38], including unhealthy snacking in university student populations [39]. During stressful situations, such as during COVID-19, young adults report higher consumption of unhealthy snacks than prior to the pandemic [40], and individuals may consume unhealthy snacks to cope with stress [41]. Due this this, the relationship between stress and unhealthy snacking can also be exacerbated by lower abilities to manage stress [42].

### 1.3. The Current Study

The aim of this study was to identify the factors of unhealthy snack consumption in a sample of young adults from Australia (aged 18 to 30 years) across the three main snack types: sweet snacks, savoury snacks and sugar-sweetened beverages. While controlling for physical activity engagement, we examined the role of intention, appetitive traits (satiety responsiveness and enjoyment of food) and stress to explain unhealthy snack consumption and bridge the intention–behaviour gap. The following was hypothesised:

**H1:** *Intention to avoid unhealthy snacking will account for significant variance in the consumption of sweet snacks, savoury snacks and sugar-sweetened beverages beyond physical activity engagement*.

**H2:** *Appetitive traits (enjoyment of food and satiety responsiveness) and stress will account for significant variance in the consumption of sweet snacks, savoury snacks and sugar-sweetened beverages beyond intention and physical activity engagement*.

## 2. Materials and Methods

### 2.1. Participants

Australian young adults aged between 18 and 30 years (*M* = 24.73, *SD* = 3.23) were recruited from September to November 2021 through snowball sampling and social media (Facebook and X, formerly known as Twitter). Respondents were eligible to participate if they were aged between 18 and 30 years, residing in Australia and not pregnant. A total of 440 responses were recorded, of which 323 were retained in the final dataset after excluding those who failed attention check questions, had missing responses for entire scales or did not meet the eligibility criteria. A sensitivity analysis conducted with G*Power 3.1 [43] indicated that our sample size of 323 participants is sufficiently powered to detect a small to moderate effect size of *f*^2^ = 0.04, with α = 0.05, power at 0.8 and six predictors for linear regressions [44]. See Table 1 for a summary of participant demographics.

### 2.2. Procedure

Participants completed a cross-sectional online questionnaire on Qualtrics, an online survey hosting website. On the first page of the survey, participants were shown a participant information sheet and required to indicate informed consent by checking a box. Those who did not indicate consent were removed from the survey and thanked for their time. Participants then completed questions determining their eligibility. Those who did not meet the eligibility criteria were automatically removed from the survey and thanked for their time. The remaining participants completed measures assessing their demographics, snacking intentions, enjoyment of food, satiety responsiveness, stress, physical activity engagement and unhealthy snacking behaviours. At the end of this study, participants were thanked for their time and asked to provide their email address so they could receive a reward. Everyone who completed the 20–25-min online survey was entered into a prize draw to win one of six AUD $50 gift cards. This amount was slightly more than what someone would earn at minimum wage for the time it took to complete the survey. Data from participants who did not provide their email address for reimbursement were still included in the final analyses. The Curtin University Human Research Ethics Committee approved this study on 21 July 2021 (HREC number RDHS-225-15).

### 2.3. Measures

Participants completed demographic questions and measures assessing their physical activity engagement, intention, appetitive traits (enjoyment of food and satiety responsiveness), stress, and consumption of the three main types of unhealthy snacks: sweet snacks, savoury snacks and sugar-sweetened beverages.

#### 2.3.1. Physical Activity Engagement

Physical activity engagement was assessed by one item (“On average, how many times per week do you engage in physical activity that gets your heart racing faster and makes you sweat for longer than 30 min?”) answered on an 8-point Likert scale (0 = never or less than once per week, 7 = daily). Only one item was used to reduce participant burden, as this study only aimed to determine physical activity engagement, rather than variations in unhealthy snacking across physical activity domains. Studies that have used single-item physical activity measures found good test-retest reliability and concurrent validity with longer self-report measures of physical activity [45,46,47]. Higher scores indicate greater frequency of engaging in physical activity per week.

#### 2.3.2. Intention

Intention to avoid unhealthy snacking was assessed by one item (“Over the next week, I will try to avoid eating unhealthy snacks”) answered on a 7-point Likert scale (1 = strongly disagree, 7 = strongly agree). This item was developed following the guidelines of Ajzen [48] to construct a theory of planned behaviour questionnaire. Only one item was used to reduce participant burden, as previous studies that had included multiple-item measures of intention indicated high intercorrelations between items [49,50]. Higher scores indicate a stronger intention to avoid consuming unhealthy snacks.

#### 2.3.3. Appetitive Traits

Appetitive traits were assessed using The Adult Eating Behaviour Questionnaire [51]. This measure assesses eight factors of appetitive traits in adulthood. However, only the enjoyment of food and satiety responsiveness factors were assessed in this study. Three items assessing enjoyment of food (e.g., “I enjoy eating”) and three items assessing satiety responsiveness (e.g., “I get full up easily”) were answered on a 5-point Likert scale (1 = strongly disagree, 5 = strongly agree). Mean scores for each subscale were calculated, with higher scores indicating greater enjoyment of food and satiety responsiveness. Internal consistency for the enjoyment of food subscale was good (α = 0.85) and acceptable (α = 0.75) for the satiety responsiveness subscale [51]. In this sample, internal consistency for the enjoyment of food subscale is good (α = 0.88) and acceptable (α = 0.78) for the satiety responsiveness subscale.

#### 2.3.4. Stress

Stress was assessed by one item (“On a scale from 1 to 10, with 1 being not stressed at all, and 10 being very stressed, how would you rate your stress in the past 30 days?”) answered on a 10-point Likert scale (1 = not stressed, 10 = very stressed). Only one item was used to reduce participant burden, as this study only aimed to screen for stress symptoms rather than conduct a comprehensive clinical assessment. Studies that have used single-item stress measures found good test-retest reliability and concurrent validity with longer self-report measures of stress [52,53]. Higher scores indicate greater stress experienced over the past 30 days.

#### 2.3.5. Unhealthy Snack Consumption

Unhealthy snack consumption was assessed by three items. Participants were provided a definition of unhealthy snacks and examples of foods that would be considered as sweet snacks, savoury snacks and sugar-sweetened beverages. They were then instructed, “Please indicate how often you consume the following snacks on average using the scales provided.” Consumption of the three unhealthy snack types were assessed by one item each, “Sweet snacks (e.g., biscuits, cakes, lollies, chocolate bars, pastries, ice-cream)”, “Salty or savoury snacks (e.g., pies, pastries, biscuits, crisps/chips, processed meats)”, and “Sugar-sweetened beverages (e.g., soft drinks, mixers for alcohol, sugar-added juices, energy drinks, hot chocolate, iced tea, cordial, fruit drinks)”. Items were responded to on a 7-point Likert scale (1 = a couple of times per year, 7 = twice or more times per day). Items were developed based on an established food frequency questionnaire [54] and the Australian Dietary Guidelines [55] to best reflect foods commonly consumed by the Australian population. Higher scores indicate greater consumption of the three unhealthy snack types.

### 2.4. Data Analysis

Data was analysed using SPSS Version 28 [56]. Descriptive statistics and bivariate correlations were assessed. Three hierarchical multiple regression analyses were then conducted to identify the factors of unhealthy snack consumption across three snack types: sweet snacks, savoury snacks and sugar-sweetened beverages. Hierarchical multiple regression analyses were applied instead of structural equation modelling due to much larger sample sizes typically required in structural equation modelling to achieve adequate statistical power and ensure stable and reliable parameter estimates [57]. Additionally, there were no planned moderation or mediation analyses, which required these pathways to be assessed via structural equation modelling [58].

## 3. Results

On average, participants consumed sweet snacks three or more times per week (*M* = 5.36, *SD* = 1.29), salty snacks once weekly (*M* = 4.82, *SD* = 1.41) and sugar-sweetened beverages once weekly (*M* = 4.03, *SD* = 1.80). See Table 2 for descriptive statistics and bivariate correlations between the variables.

### 3.1. Identifying the Factors of Sweet Snack Consumption

In step one, physical activity engagement was added to the model as a control variable. In step two, intention was added to the model. In step three, enjoyment of food and satiety responsiveness were added to the model. In the final step, stress was added to the model. The overall model accounted for a significant 6.4% of variance in sweet snack consumption, *R^2^* = 0.06, *F* (5, 316) = 4.31, *p* < 0.001. Physical activity engagement and stress were the only significant factors. See Table 3 for the individual contributions of variables in each step of the model.

### 3.2. Identifying the Factors of Savoury Snack Consumption

In step one, physical activity engagement was added to the model as a control variable. In step two, intention was added to the model. In step three, enjoyment of food and satiety responsiveness were added to the model. In the final step, stress was added to the model. The overall model accounted for a significant 6.6% of variance in savoury snack consumption, *R^2^* = 0.07, *F* (5, 316) = 4.48, *p* < 0.001. Physical activity engagement was the only significant factor. See Table 4 for the individual contributions of variables in each step of the model.

### 3.3. Identifying the Factors of Sugar-Sweetened Beverage Consumption

In step one, physical activity engagement was added to the model as a control variable. In step two, intention was added to the model. In step three, enjoyment of food and satiety responsiveness were added to the model. In the final step, stress was added to the model. The overall model accounted for a significant 6.2% of variance in sugar-sweetened beverage consumption, *R^2^* = 0.06, *F* (5, 316) = 4.19, *p* < 0.001. Physical activity engagement and satiety responsiveness were the only significant factors. See Table 5 for the individual contributions of variables in each step of the model.

## 4. Discussion

The aim of this study was to identify the factors of unhealthy snack consumption among young adults (aged 18 to 30 years) across the three main snack types: sweet snacks, savoury snacks and sugar-sweetened beverages. While controlling for physical activity engagement, we examined the role of intention, appetitive traits (satiety responsiveness and enjoyment of food) and stress to explain unhealthy snack consumption and bridge the intention–behaviour gap. The findings do not support Hypothesis 1, as intention was not a significant factor for any snack type and did not account for unique variance beyond physical activity engagement. However, the findings provide partial support for Hypothesis 2. Specifically, physical activity engagement and stress were significant factors in sweet snack consumption, with stress accounting for unique variance beyond physical activity engagement and intention. However, physical activity engagement was the only significant factor of savoury snack consumption, and no other factors could account for unique variance beyond physical activity engagement. For sugar-sweetened beverage consumption, physical activity engagement and satiety responsiveness were significant factors, with satiety responsiveness accounting for unique variance beyond physical activity engagement and intention.

Physical activity engagement was a significant factor for all three snack types. Findings support prior suggestions to consider physical activity as a confounding factor of unhealthy snacking [7] but contradict findings indicating that exercise did not significantly predict snacking frequency [59]. Similarly, current results indicate a negative relationship between physical activity engagement and unhealthy snacking. This contrasts with prior research that found individuals are more likely to consume snacks when they have perceived greater energy expenditure [33]. It could be that individuals who frequently engage in exercise are driven to eat more but can better regulate their appetite to balance energy intake and expenditure [60]. Therefore, the current sample, consisting of those who typically engage in physical activity several times per week (*M* = 2.53, *SD* = 2.05), may reflect this. Due to the contradictory findings in the literature, it may be useful to instead explore the psychological or physiological processes underlying the link between physical activity and unhealthy snacking. For example, future research may consider the role of compensatory beliefs (e.g., believing that engaging in healthy behaviours can offset the negative effects of unhealthy behaviours) to resolve the cognitive dissonance between wanting to be healthy and wanting to eat snacks [61]. Alternatively, snack consumption may be considered a physiological response to expending energy when exercising [7] or a mechanism to regulate appetite [60].

Intention was not a significant factor in the consumption of any snack type. The findings contradict prior research, which found that intention was an important predictor of snacking behaviour [50,62,63]. Additionally, the findings contradict the theory of planned behaviour [20] and recent meta-analytic findings, which indicate that intention is the largest predictor of dietary behaviours [64]. The findings may be attributed to this study’s sample, which consisted of young adults, as relationships between intention and dietary behaviours tend to be stronger for older, compared to younger individuals [64]. Alternatively, the findings may be attributed to the avoidance-framed intention measure used in this study. Specifically, McAlpine, Mullan [65] found that approach-framed intentions (e.g., “I intend to ___”) are generally stronger predictors of behaviour than avoidance-framed intentions (e.g., “I intend to avoid ___”). Similarly, intention–behaviour relationships are generally weaker for behaviours that individuals often aim to avoid, such as unhealthy snacking [66]. This is because non-conscious processes may override intentions when inhibiting the dominant automatic urges typically experienced in hedonically driven behaviours [67]. Therefore, it may be difficult to accurately report momentary intentions to avoid unhealthy dietary behaviours with long-term risks but immediate rewards [50]. Consequently, future research should consider approach-framed intention measures to better capture momentary unhealthy snacking intentions or consider more automatic or habitual mechanisms to better understand the factors of unhealthy snacking.

Enjoyment of food was not a significant factor in the consumption of any snack type. The influence of the enjoyment of food on snacking behaviours is rarely assessed in young adult populations. However, the findings contradict prior research indicating that greater enjoyment of food in childhood is associated with greater appetites and overweight or obesity in later life [35]. Similarly, findings contradict the results of an experiment, which aimed to increase participants’ remembered enjoyment of food, and found this manipulation to increase consumption [68]. However, the current research assessed the enjoyment of food in young adults and measured enjoyment of food as an appetitive trait. Therefore, the observed impacts may differ from those found in childhood assessments of appetitive traits or the experimental changes to state enjoyment of food. Future research may investigate whether there is a link between trait enjoyment of food in young adults and unhealthy snacking to clarify the evidence.

Satiety responsiveness was a significant factor of sugar-sweetened beverage consumption, but not sweet or savoury snack consumption. Findings contradict prior research indicating that low satiety responsiveness was associated with overall greater energy consumption [69,70]. However, the current study differentiated the various snack types to highlight that satiety responsiveness was only a significant factor of sugar-sweetened beverage consumption, and that higher satiety responsiveness influenced greater sugar-sweetened beverage consumption. Liquid and low-viscosity foods are less likely to result in fullness compared to solid and high-viscosity foods [71]. Therefore, it is possible that individuals who are highly attuned to satiety signals and rely on this to regulate their diets may still consume sugar-sweetened beverages, as these beverages may bypass the mechanisms which signal fullness [72]. To reduce sugar-sweetened beverage consumption in individuals with high satiety responsiveness, interventions may encourage the consumption of foods that generate strong satiety sensations. This includes foods high in protein, carbohydrate or fibre macronutrients, or foods high in texture [73].

Stress was a significant factor in sweet snack consumption, but not in savoury snack or sugar-sweetened beverage consumption. Findings are partially consistent with recent research indicating that young adults are more likely to consume all snack types during stressful periods, such as COVID-19 [40] and results of a scoping review, which found that individuals experiencing high levels of stress most preferred energy-dense snacks, followed by sweet, savoury and healthy options [74]. However, Mohamed, Mahfouz [75] found gender differences in snack type preferences among young adults experiencing high levels of stress, where women are more likely to prefer sweet snacks and men are more likely to prefer savoury snacks. The current study consisted mainly of women (94.1%), so the current findings may instead reflect the factors of unhealthy snacking in young women. Therefore, further research may consider recruiting more diverse samples to better understand the factors of unhealthy snacking in the general young adult population.

### Strengths and Limitations

This study provides evidence for the differential importance of the psychological factors contributing to the consumption of various snack types amongst young adults. Prior research has considered the psychological factors of general unhealthy snack consumption, e.g., [69,70,76]. However, this study has differentiated the various types of unhealthy snacks to more accurately assess the factors contributing to the consumption of each main snack type. Therefore, policymakers, healthcare practitioners or researchers aiming to design interventions to reduce unhealthy snacking may consider the study findings to tailor the intervention methods to effectively target the influential factors of each snack type.

The current study has also identified the psychological factors that could bridge the intention–behaviour gap by considering both intentional and non-intentional factors of unhealthy snacking. Contrary to previous meta-analytic findings [64], the current results suggest that non-conscious or automatic factors, rather than rationally driven intentions, may be more influential in unhealthy snack consumption amongst young adults. Although findings may be attributed to the framing of the intention measure in this study, findings highlight key theoretical and measurement considerations to improve the conceptualisation of intention measures.

However, it should be noted that participants self-reported their consumption of unhealthy snacks in this study, and it is possible that individuals may underreport their dietary consumption or experience difficulties related to memory recall and estimating quantities [77]. Steps were taken to minimise the risk of self-report errors by providing examples of food items under each snack type. But future research may use alternative assessment methods when more accurate dietary records are required (e.g., via 24-h dietary recall or ecological momentary assessment methods to improve memory recall, or observational studies to improve the accuracy of food quantities recorded).

Similarly, stress was self-reported over a 30-day period, and this data may be impacted by recall biases, including potential overestimation or underestimation of stress levels, recent events, mood at the time of reporting and individual differences in cognitive bias [78]. Although stress measures are most commonly self-reported, findings should be interpreted considering these limitations and differentiated from objective measures of physiological stress [79].

Additionally, it should be noted that data were collected during COVID-19, when some participants indicated that they were experiencing COVID-19 restrictions. For these individuals, it is possible that they may have experienced changes in unhealthy food and drink consumption [80], psychological states [81] and physical activity [82] during the data collection period. Notably, individuals’ decreased psychological well-being during COVID-19 [83] may influence the generalisability of results in a non-pandemic context. Nevertheless, this study provides valuable insights into the experiences of young adults in unprecedented global crises, which may be transferable to enhance future public health responses.

Similarly, this study has employed a cross-sectional design so that longitudinal effects cannot be inferred [84]. Additionally, the current sample mostly consists of women, so that gender differences cannot be inferred, and the current sample may impact the generalisability of findings to the general young adult population. However, this demographic is essential to understanding snacking behaviours in young adults, as young women frequently report high engagement in dieting, emotional eating and body image concerns, which may uniquely influence dietary behaviours [85].

## 5. Conclusions

Differences in the importance of psychological factors across unhealthy snack types were found and should be considered in tailored health behaviour change interventions to reduce the consumption of various snack types. Physical activity engagement and stress were significant factors of sweet snack consumption. Physical activity engagement was the only significant factor of savoury snack consumption. Physical activity engagement and satiety responsiveness were significant factors of sugar-sweetened beverage consumption. Findings indicate the psychological factors that could bridge the intention–behaviour gap in unhealthy snacking by considering both intentional and non-intentional processes of behaviour. Rational dietary decision-making can be interrupted by less conscious cognitive or physiological processes. Therefore, health practitioners and health promotion campaigns may consider the involvement of psychologists to mitigate the influence of less conscious psychological factors to encourage healthy eating patterns.

## Figures and Tables

**Table 1 nutrients-17-02681-t001:** Participant demographics (*N* = 323).

	*N*	%
**Gender**		
Woman	304	94.1
Man	19	5.9
**Australian state of residence**		
Western Australia	155	48.0
Victoria	54	16.7
New South Wales	51	15.8
Queensland	30	9.3
Australian Capital Territory	14	4.3
South Australia	14	4.3
Tasmania	5	1.5
**Physical activity engagement**		
Never/less than once per week	58	18.0
One day	65	20.2
Two days	57	17.7
Three days	48	14.9
Four days	33	10.2
Five days	24	7.5
Six days	20	6.2
Daily	17	5.3
**COVID-19 restrictions**		
Yes	122	37.8
No	201	62.2
**COVID-19 measures (*N* = 122) ***		
Full lockdown	91	74.6
Some restrictions	31	25.4

Note. * Assessed only from the proportion of respondents who were experiencing COVID-19 restrictions at the time of data collection.

**Table 2 nutrients-17-02681-t002:** Descriptive statistics and bivariate correlations between variables (*N* = 323).

	*M* (*SD*)	1.	2.	3.	4.	5.	6.	7.	8.	9.	10.
1. Gender	-	-	0.03	0.11	0.06	0.00	−0.04	−0.15 **	−0.20 **	−0.15 **	−0.08
2. Age	24.73 (3.23)		-	0.06	−0.06	0.19 **	0.01	−0.07	−0.11 *	−0.09	−0.06
3. Physical activity engagement	2.53 (2.05)			-	0.03	−0.05	−0.00	−0.10	−0.19 **	−0.22 **	−0.15 **
4. Intention to avoid unhealthy snacking	4.90 (1.53)				-	0.04	0.02	−0.01	−0.07	−0.08	0.02
5. Enjoyment of food	4.24 (0.81)					-	−0.31 **	−0.10	−0.04	0.01	−0.04
6. Satiety responsiveness	2.81 (0.90)						-	0.07	0.09	0.07	0.19 **
7. Stress	7.06 (2.11)							-	0.15 **	0.10	0.11 **
8. Sweet snack consumption	5.36 (1.29)								-	0.48 **	0.31 **
9. Savoury snack consumption	4.82 (1.41)									-	0.24 **
10. Sugar-sweetened beverage consumption	4.03 (1.80)										-

Note. * *p* < 0.05, ** *p* < 0.001.

**Table 3 nutrients-17-02681-t003:** Individual contributions of variables in the regression explaining sweet snack consumption (*N* = 323).

Step	Predictor	*B* [95% CI]	SE	β	sr^2^	*R* ^2^	Δ*R*^2^	*F* (*df*)	Δ*F* (*df*)
1	Physical activity engagement	−0.12 [−0.19, −0.05] **	0.04	−0.19	−0.04	0.04	0.04	11.86 (1, 320)	11.86 (1, 320)
2	Physical activity engagement	−0.12 [−0.19, −0.05] **	0.04	−0.19	−0.03	0.04	0.00	6.58 (2, 319)	1.29 (1, 319)
	Intention	−0.05 [−0.14, 0.04]	0.05	−0.06	−0.00				
3	Physical activity engagement	−0.12 [−0.19, −0.05] **	0.04	−0.19	−0.04	0.05	0.01	4.06 (4, 317)	1.52 (2, 317)
	Intention	−0.05 [−0.14, 0.04]	0.05	−0.06	−0.00				
	Enjoyment of food	−0.03 [−0.21, 0.16]	0.09	−0.02	−0.00				
	Satiety responsiveness	0.13 [−0.04, 0.29]	0.08	0.09	0.01				
4	Physical activity engagement	−0.11 [−0.18, −0.04] *	0.04	−0.18	−0.03	0.06	0.02	4.31 (5, 316)	5.11 (1, 316)
	Intention	−0.05 [−0.14, 0.04]	0.05	−0.06	−0.00				
	Enjoyment of food	−0.01 [−0.19, 0.17]	0.09	−0.01	−0.00				
	Satiety responsiveness	0.12 [−0.04, 0.28]	0.08	0.08	0.00				
	Stress	0.08 [0.10, 0.14] *	0.03	0.12	0.02				

Note. * *p* < 0.05, ** *p* < 0.001.

**Table 4 nutrients-17-02681-t004:** Individual contributions of variables in the regression explaining savoury snack consumption (*N* = 323).

Step	Predictor	*B* [95% CI]	SE	β	sr^2^	*R* ^2^	Δ*R*^2^	*F* (*df*)	Δ*F* (*df*)
1	Physical activity engagement	−0.15 [−0.23, −0.08] **	0.04	−0.22	−0.05	0.05	0.05	16.18 (1, 320)	16.18 (1, 320)
2	Physical activity engagement	−0.15 [−0.22, −0.08] **	0.04	−0.22	−0.05	0.05	0.01	8.88 (2, 319)	1.55 (1, 319)
	Intention	−0.06 [−0.16, 0.04]	0.05	−0.07	−0.00				
3	Physical activity engagement	−0.15 [−0.22, −0.08] **	0.04	−0.22	−0.05	0.06	0.01	5.05 (4, 317)	1.22 (2, 317)
	Intention	−0.07 [−0.16, 0.03]	0.05	−0.07	−0.00				
	Enjoyment of food	0.04 [−0.16, 0.24]	0.10	0.02	0.00				
	Satiety responsiveness	0.14 [−0.04, 0.32]	0.09	0.09	0.01				
4	Physical activity engagement	−0.14 [−0.22, −0.07] **	0.04	−0.21	−0.04	0.07	0.01	4.48 (5, 316)	2.11 (1, 316)
	Intention	−0.06 [−0.16, 0.03]	0.05	−0.07	−0.00				
	Enjoyment of food	0.05 [−0.14, 0.25]	0.10	0.03	0.00				
	Satiety responsiveness	0.14 [−0.04, 0.31]	0.09	0.09	0.01				
	Stress	0.05 [−0.02, 0.13]	0.04	0.08	0.01				

Note. ** *p* < 0.001.

**Table 5 nutrients-17-02681-t005:** Individual contributions of variables in the regression explaining sugar-sweetened beverage consumption (*N* = 323).

Step	Predictor	*B* [95% CI]	SE	β	sr^2^	*R* ^2^	Δ*R*^2^	*F* (*df*)	Δ*F* (*df*)
1	Physical activity engagement	−0.13 [−0.23, −0.03] *	0.05	−0.15	−0.02	0.02	0.02	7.13 (1, 320)	7.13 (1, 320)
2	Physical activity engagement	−0.13 [−0.23, −0.03] *	0.05	−0.15	−0.02	0.02	0.00	3.59 (2, 319)	0.07 (1, 319)
	Intention	0.02 [−0.11, 0.15]	0.07	0.02	0.00				
3	Physical activity engagement	−0.13 [−0.22, −0.03] *	0.05	−0.15	−0.02	0.06	0.03	4.59 (4, 317)	5.49 (2, 317)
	Intention	0.01 [−0.11, 0.15]	0.07	0.01	0.00				
	Enjoyment of food	0.04 [−0.22, 0.29]	0.13	0.02	0.00				
	Satiety responsiveness	0.37 [0.14, 0.60] *	0.12	0.19	0.04				
4	Physical activity engagement	−0.12 [−0.22, −0.03] *	0.05	−0.14	−0.02	0.06	0.01	4.19 (5, 316)	2.49 (1, 316)
	Intention	0.01 [−0.11, 0.14]	0.06	0.01	0.00				
	Enjoyment of food	0.05 [−0.20, 0.30]	0.13	0.02	0.00				
	Satiety responsiveness	0.36 [0.14, 0.59] *	0.12	0.18	0.03				
	Stress	0.07 [−0.02, 0.16]	0.05	0.09	0.01				

Note. * *p* < 0.05.

## Data Availability

The data presented in this study are available on request from the corresponding author due to privacy and ethical restrictions.

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
