# Peer review of "The Role of Psychological Factors in Young Adult Snacking: Exploring the Intention–Behaviour Gap"

_nutrients, 2025, doi:10.3390/nu17162681_

Round 1

Reviewer 1 Report

Comments and Suggestions for Authors

  1. 'High consumption of unhealthy snacks, or 45 foods high in sugar, salt and fats, and low in nutrients [1] are associated with adverse 46 health implications. These include the development of chronic and non-communicable 47 diseases such as cardiovascular diseases, Type 2 diabetes, cancers, obesity and depression 48 [1, 7-12].'   are there any differences between male and female and in a developmental fashion? could the author address this?
  2. 'This is known as the intention-behaviour gap and is commonly noted as a weakness 77 of the theory of planned behaviour [25, 26]. In subsequent attempts to improve the pre-78 dictive ability of the theory of planned behaviour, Hall and Fong [26] note that the theory 79 may best predict conscious, controllable and less habitual behaviours.'  again, maybe cite here which variables may be of interest

  3. '

    Participants were aged between 18 and 30 years (M = 24.73, SD = 3.23) and most iden-127 tified as women (94.1%) while 5.9% identified as men. Most resided in Western Australia 128 (48%), while the remainder lived in Victoria (16.7%), New South Wales (15.8%), Queens-129 land (9.3%) Australian Capital Territory (4.3%), South Australia (4.3%) or Tasmania 130 (1.5%). During the data collection period, most (74.6%) indicated that they were currently 131 in a full COVID-19 lockdown period. Most participants (70.9%) also had a household in-132 come below $104,999, while the remainder (29.9%) had a household income between 133 $105,000 - $200,000 or greater. Almost half of all participants (48.6%) had a BMI measure 134 in the ‘healthy weight’ range between 18.5 to 24.9 (M = 26.14, SD = 6.48), did not experience.

    anxiety or depression (66.6%) and engaged in physical activity 2 to 3 days a week (M = 136 2.53, SD = 2.05).

    .' this reviewer thinks that alla this information could be summarized in a a table, particularly data on BMI and Psychological variables.

    How were data about anxiety or depression collected? questionaires? single items response?
  4. statistical analyses and data results. Although the statistical method is valid and useful this section and consequently the results section is very hard to follow. I would suggest to use tables to describe the results.
  5. Maybe to let the readers better follow thee statistical reasoning it would be easier to use a SEM model such as a moderation model rather than different logistic regressions.
  6. the discussion should be modified if SEM is adopted.

thank you for the opportunity to revise this ms. the topic is really important and useful

Reviewer 2 Report

Comments and Suggestions for Authors

Comments and Suggestions for Authors

Manuscript Nutrients-3768454

The objective of this study was to examine the determinants of unhealthy snack consumption among young adults (aged 18 to 30 years), with a focus on three major snack categories: sweet snacks, savory snacks, and sugar-sweetened beverages. The findings indicate that the psychological factors influencing consumption vary across snack types, highlighting the need for snack-type-specific interventions. Engagement in physical activity and perceived stress emerged as significant predictors of sweet snack consumption, whereas only physical activity engagement was significantly associated with savory snack consumption. Given its methodological rigor and practical implications for nutritional intervention strategies, this study presents valuable insights and may be considered for publication in Nutrients.

Below, I include some suggestions for the authors to improve the quality of the manuscript.

SUGGESTIONS

The title, summary, and keywords are appropriate.

INTRODUCTION

The introduction is clearly written and well-structured, albeit somewhat lengthy—understandably so, given the inclusion of three minor sub-sections. Nevertheless, it would benefit from a more thorough integration of recent, relevant literature to strengthen its contextual foundation.

MATERIAL AND METHODS

Clarified Version: At the end of the study, participants were thanked for their time and asked to provide their contact details so they could receive a reward. Everyone who completed the 20–25-minute online survey was entered into a prize draw to win one of six AUD $50 gift cards. This amount was slightly more than what someone would earn at minimum wage for the time it took to complete the survey.

As for how participants were excluded: could you clarify whether respondents were automatically screened out during the survey if they didn’t meet the required criteria (e.g., based on their answers to early questions), or whether all responses were collected first and then filtered later during data cleaning and processing?

Stress was assessed using a single-item measure: “On a scale from 1 to 10, with 1 being not stressed at all and 10 being extremely stressed, how would you rate your stress over the past 30 days?” Responses were recorded on a 10-point Likert-type scale (1 = not stressed, 10 = very stressed). A single-item approach was chosen to minimize participant burden, as the aim was to screen for general stress levels rather than conduct a detailed clinical evaluation. Prior research has demonstrated that single-item stress measures can yield acceptable levels of test-retest reliability and show strong concurrent validity with multi-item self-report instruments [44, 45]. Higher scores reflect greater perceived stress during the preceding 30 days.

However, it should be noted that retrospective self-assessment of stress over a 30-day period may be subject to recall biases, including potential overestimation or underestimation of stress levels. This limitation should be considered when interpreting the results, as memory-based judgments of stress can be influenced by recent events, mood at the time of reporting, and individual differences in cognitive bias.

Unhealthy snack consumption was assessed using a modified version of the Swiss Snack Consumption Scale. However, it remains unclear whether this modified scale has been previously validated on an Australian sample.

The research methods are thoroughly described and appropriately justified. The sampling strategy, measurement instruments, and analytical procedures are suitable and rigorously applied. Ethical standards have been duly observed throughout the study. The data collected demonstrate validity and reliability.

RESULTS

Results are presented with clarity and interpreted accurately in line with the research objectives.

DISCUSSION

The discussion would benefit from a clearer scientific grounding by integrating and contextualizing recent findings from similar studies, thereby better connecting them with the current research.

CONCLUSION

While the conclusions are substantiated by the data, I suggest they be revised in light of the recent modifications.

Recommendation:

Accept with minor changes

Reviewer 3 Report

Comments and Suggestions for Authors

In the abstract, add the mean age (and standard deviation) of the respondents.

Line 106: among young adults (18 to 30 years) – please add: "in a sample of young adults from Australia".

Lines 132-135: What is the purpose of describing participants based on income and BMI if the results are not presented according to these variables? Please remove.

Line 136: What is the purpose of describing participants based on anxiety and depression if the results are not presented according to these variables? Furthermore, the method of assessing these variables is not specified. Please remove.

Line 136: engaged in physical activity 2 to 3 days a week (M =2.53,SD=2.05). Please add the %.

Add inclusion and exclusion criteria for the participants, instead of ‘‘A total of 440 responses were recorded, of which 323 were retained in the final dataset after excluding those who failed attention check questions, missing responses for entire scales, are pregnant or not residing in Australia‘‘.

Line 143-145: ~Participants then completed questions determining their eligibility. Those who did 143 not meet the eligibility criteria (i.e., those not aged between 18-30 years, were pregnant or 144 did not reside in Australia) were removed from the survey and thanked for their time. Participants who met all eligibility requirements~. Please remove.

Line 149: their contact details for reimbursement. What specific contact details were collected? And if participants did not agree to provide them, were they still included in the study?

Line 200:  Modified Swiss Food Panel Food Frequency Questionnaire [46]. The qustionanaire is validated in English? Please explain.

Discussion: Discuss the results also in light of the hypotheses formulated: whether they were confirmed, refuted, or partially supported, etc.

Conclusions: ,,The aim of this study was to identify the factors of unhealthy snack consumption 401 among young adults (18 to 30 years) across the three main snack types: sweet snacks, sa-402 voury snacks and sugar-sweetened beverages. We examined the role of physical activity 403 engagement as a control variable, and assessed the influence of intention, appetitive traits 404 (satiety responsiveness and enjoyment of food) and stress to explain unhealthy snack con-405 sumption and bridge the intention-behaviour gap~. The pragraph is repetitive, please remove.

To be added to the conclusions: the practical implications of the obtained results, such as the involvement of psychologists alongside medical professionals in health education, the promotion of a healthy diet, lifestyle, and so on.

The title should mention that the study was conducted during the COVID-19 lockdown. In the discussion section, it should be emphasized that during the pandemic, other factors — especially psychological ones — may have also interfered or influenced the results.

Round 2

Reviewer 3 Report

Comments and Suggestions for Authors

Dear authors,

All the comments were addressed, thank you and good luck.